# EXPLAIN, EDIT, GENERATE: Rationale-Sensitive Counterfactual Data Augmentation for Multi-hop Fact Verification

**Yingjie Zhu**[1*], **Jiasheng Si**[1*], **Yibo Zhao**[1], **Haiyang Zhu**[1], **Deyu Zhou**[1†], **Yulan He**[2,3]

[1]School of Computer Science and Engineering, Key Laboratory of Computer Network and Information Integration, Ministry of Education, Southeast University, China
[2] Department of Informatics, King's College London, UK
[3] The Alan Turing Institute, UK
{yj_zhu, jasenchn, yibozhao, haiyangzhu, d.zhou}@seu.edu.cn
yulan.he@kcl.ac.uk

## Abstract

Automatic multi-hop fact verification task has gained significant attention in recent years. Despite impressive results, these well-designed models perform poorly on out-of-domain data. One possible solution is to augment the training data with counterfactuals, which are generated by minimally altering the causal features of the original data. However, current counterfactual data augmentation techniques fail to handle multi-hop fact verification due to their incapability to preserve the complex logical relationships within multiple correlated texts. In this paper, we overcome this limitation by developing a rationale-sensitive method to generate *linguistically diverse* and *label-flipping* counterfactuals while preserving *logical relationships*. In specific, the diverse and fluent counterfactuals are generated via an Explain-Edit-Generate architecture. Moreover, the checking and filtering modules are proposed to regularize the counterfactual data with logical relations and flipped labels. Experimental results show that the proposed approach outperforms the SOTA baselines and can generate linguistically diverse counterfactual data without disrupting their logical relationships[1].

## 1 Introduction

Multi-hop fact verification task, which discerns the truth from falsehood based on multiple hops of reliable evidence, becomes crucial in countering misinformation and counterfeit news spread on current social media platforms (Vosoughi et al., 2018; Botnevik et al., 2020), especially in some specific domains such as politics (Alhindi et al., 2018; Ostrowski et al., 2021), science (Wadden et al., 2020, 2022) and public health (Kotonya and Toni, 2020; Sarrouti et al., 2021). However, many recent works often perform poorly under the multitude of distribution shifts due to an over-reliance on spurious correlations between input text and labels (Gururangan et al., 2018; Schuster et al., 2019; Geirhos et al., 2020). It can potentially be addressed by Counterfactual Data Augmentation (CDA), using counterfactual instances generated by perturbing causal features within the input (Khashabi et al., 2020). Several works have revealed that training with counterfactual data enhances the capability of the model to identify causal features and diminish its reliance on spurious correlations between the input text and the label, thus resulting in the improvement in Out-Of-Domain (OOD) generalization (Vig et al., 2020; Eisenstein, 2022).

In this paper, we seek to generate counterfactuals for multi-hop fact verification, instead of exploring the causal bias for a specific model. However, due to the complex logical relationships within the multi-hop input texts, developing such an approach poses some significant challenges. As shown in the first row of Table 1, most CDA methods are designed for NLP tasks without requiring intricate reasoning over the input, such as the sentiment analysis task (Yang et al., 2021; Howard et al., 2022). Their local modification of the causal feature in a single sentence (e.g., *"amazing"* in Table 1 ⇒ *"terrible"*) is difficult to constrain the *logical relationships* between different causal features in multiple correlated texts, resulting in unverifiable counterfactuals. Furthermore, the prior attempt, CrossAug (Lee et al., 2021), is primarily designed to generate counterfactuals for single-hop fact verification via consistently editing the causal features in the claim and in the one piece of evidence (e.g., *"over 30 days "* in the second row of Table 1 ⇒ *"less than 10 days"*). Nevertheless, its claim-only based generation strategy struggles to preserve the complex logical relationships when faced with multiple hops of evidence, and fails to ensure *label flipping* and *linguistic diversity* in the counterfactuals, which

---

*Equal Contribution.
†Corresponding Author.
[1]The code and datasets are available at https://github.com/AAAndy-Zhu/RACE

| Task | Inference | Input X (label Y) |
|---|---|---|
| Sentiment Analysis | X → Y | This is an amazing book, I'm already immersed in the storyline. **(POSITIVE)** |
| Single-hop Fact Verification | | **C:** Little Miss Sunshine was filmed over 30 days. **(SUPPORTS)** |
| | | **E:** Little Miss Sunshine ..., filming began on June and took place over 30 days in Arizona ... |
| Multi-hop Fact Verification | Reasoning Graph | **C:** The Ford Fusion was introduced for model year 2006. The Rookie of The Year in the 1997 CART season drives it in the NASCAR Sprint Cup Series. **(SUPPORTS)** |
| | | **E1:** Ford Fusion is manufactured and marketed by Ford. Introduced for the 2006 model year, ... |
| | | **E2:** Patrick Carpentier competed in the NASCAR Sprint Cup Series, driving the Ford Fusion. |
| | | **E3:** The 1997 CART PPG World Series season, ... Rookie of the Year was Patrick Carpentier. |

Table 1: Comparison between different tasks.

are crucial for CDA (Joshi and He, 2022).

For multi-hop fact verification, as shown in the third row of Table 1, the set of possible causal features is more complex, and exploring them may necessitate intricate reasoning about the logical relationships between multiple hops of evidence and between the claim and the evidence. For example, the "*Patrick Carpentier*" in $E2$, which is invisible to the claim, bridges the connection between the causal features "*Introduced for the 2006 model year*" in $E1$ and "*Rookie of the Year*" in $E3$, thus leading to the alignment of the multi-hop evidence with the claim $C$ (as shown in the Reasoning Graph). Without considering such complex *logical relationships* within the correlated input, the generated counterfactual claims potentially tend to be unreasonable or unverified. Furthermore, ensuring the *label flipping* and *linguistic diversity* of generated counterfactuals become increasingly difficult with the premise of *logical relationships*, which are critical factors to assure the quality of the counterfactuals.

To address these challenges, we develop a novel pipeline method, RACE (**RA**tionale-sensitive **C**ounterfactual g**E**neration), by focusing on the causal features within the rationales extracted from the multi-hop evidence using an explainability method. In specific, for each original instance, the *Explainer* and *Editor* modules are employed to produce the counterfactual evidence that logically corresponds to — but factually distinct from — the original claim. Then, according to the counterfactual evidence, an entity-aware *Generator* generates the counterfactual claims by synthesizing the semantic information across multi-hop evidence. During the above process, the Checking and Filtering modules are used to regularize the reasonableness of the output of each module from different aspects, resulting in fully labeled examples that can be used directly to augment the training data. The

**motivation** here is that these rationales provide the intrinsic semantic and relational information for inferring its label, and present the factual consistency with its claim (Raha et al., 2023).

It should be pointed out that RACE requires no external knowledge as used in Paranjape et al. (2022) besides the original training data, and is able to generate *linguistically diverse* and *label-flipping* counterfactuals while preserving *logical relationships*. Compared to alternative approaches (e.g., ChatGPT (OpenAI, 2022)) (§ 4), training on the counterfactuals generated by RACE reveals the improvement in performance under different settings (§ 5.1), including in-domain, out-of-domain (Paranjape et al., 2022), and challenge settings (Gardner et al., 2020). In addition, the intrinsic evaluation shows that the counterfactual claims generated by RACE are more logical and linguistically diverse than those produced by the baselines (§ 5.3, § 5.4). Finally, we compare the results based on different generation models with baselines, illustrating that our method is generation model-agnostic (§ 5.5).

## 2 Related Works

**Debiasing Fact Verification** A variety of advanced multi-hop fact verification methods have recently emerged in various domains due to the development of pre-trained models (Das et al., 2023). Nevertheless, most models exhibit poor OOD generalization, primarily due to their over-reliance on spurious correlations between inputs and labels (Gururangan et al., 2018; Schuster et al., 2019; Geirhos et al., 2020). Thus, several works focus on the debiasing of fact verification models. Schuster et al. (2019) have identified strong cues for predicting labels solely based on the claim. Zhu et al. (2022) proposed an entity debiasing framework that mitigates entity bias from a cause-effect perspective. Lee et al. (2021) addressed the debiasing of fact verification models by augmenting the data

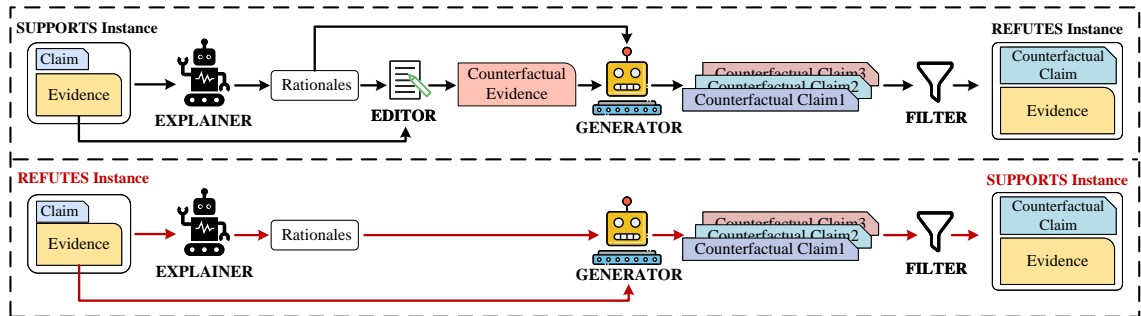

Figure 1: The overall pipeline of RACE. The *SUPPORTS* and *REFUTES* instances are processed differently, as indicated by the black and red arrows, respectively.

with contrastive instances. Atanasova et al. (2022) explored what information is sufficient to verify a claim, and proposed a CDA schema for learning of (in)sufficient information.

**Counterfactual Data Augmentation** There is a growing academic interest in CDA to improve model robustness. Initial studies focus on human-crafted counterfactuals (Kaushik et al., 2020; Gardner et al., 2020). Recently, numerous automatic CDA methods have been proposed for sentiment analysis (Wang and Culotta, 2021; Yang et al., 2021; Howard et al., 2022), question answering (Paranjape et al., 2022; Dixit et al., 2022), and natural language inference (Glockner et al., 2018). However, these methods are primarily targeted to NLP tasks without requiring complex reasoning about the input. Thus, their direct application to the multi-hop fact verification task presents considerable challenges.

## 3 Methodology

Given a claim $c$ with its associated evidence $E = (e_1, e_2, \ldots, e_n)$, the aim of multi-hop fact verification is to infer whether the claim is **supported** or **refuted** by the evidence. We denote an instance in the dataset $D$ as a triplet $(c, E, y)$, where $y \in \{SUP, REF\}$ is the verification label. The goal of RACE is to generate counterfactual data $(c', E, y')$ or $(c, E', y')$ that differ in some meaningful way from the original instance $(c, E, y)$, where $y' \neq y$, $c'$ and $E'$ denote the counterfactual claim and counterfactual evidence, respectively. This setting poses some unique challenges, such as requiring to identify the causal features to be edited, ensuring sound logical relations in evidence editing and claim generation, and avoiding unverifiable claims. Meanwhile, ensuring the semantic diversity and the minimal perturbation of the counterfactuals

can also be challenging. To this end, we propose a general pipeline, RACE, to tackle these challenges.

As shown in Figure 1, our RACE consists of four stages: (I) Explainer: rationale extraction (§3.1), (II) Editor: evidence editing (§3.2), (III) Generator: claim generation (§3.3), (IV) Filtering (§3.4). Note that our method handles $SUP$ and $REF$ instances differently, as the large difference in generation space between these two types of instances.

### 3.1 Explainer: Rationale Extraction

Our RACE focuses on identifying the causal features within rationales that can be perturbed. To this end, we use CURE (Si et al., 2023a), a multi-granular rationale extraction method, to simultaneously extract sentence rationales $R_s$ and token rationales $R_t$ from the multi-hop evidence $E$ for both $SUP$ and $REF$ instances. In essence, the token rationales $R_t$ reflect the logical correlation within the evidence (blue words in Table 1) and the factual relationship between the claim and the evidence (red words in Table 1). Considering the causal relationship of the rationales to the prediction label (Wu et al., 2022), we regard the extracted rationales as the **causal features** that are to be further processed. The detailed algorithm can be found in Si et al. (2023a).

### 3.2 Editor: Evidence Editing

In general, entities contained within the multi-hop evidence possess a rich trove of factual knowledge and crucial information (e.g., *date*, *location*, *organization*, *person*, and the correlation between them), facilitating more precise multi-hop fact verification (de Jong et al., 2021; Rani et al., 2023). Therefore, we meticulously design a set of simple entity-based evidence editing rules to control the semantic perturbation while preserving the multi-hop correlation within the evidence, and an Ad-Checking mod-

ule to filter out the under-edited or over-edited evidence. Additionally, Tan et al. (2023) highlight that controlling the generation for $REF$ is more challenging due to its significantly broader generation scope compared to $SUP$. As such, we focus on editing the evidence $E$ for instances $(c, E, SUP)$ rather than for instances $(c, E, REF)$.

**Editing** We first utilize an off-the-shelf NER tool, Stanza (Qi et al., 2020), to identify various types of **causal entity** $T$ from token rationales $R_t$. Following Rani et al. (2023), we only retain entities with specific types, including ORG, PERSON, DATE, GPE, and NUM. Then, we automatically edit the evidence according to the following rules.

    • **in-Dataset**: Randomly *replace* entities of type GPE, DATE and NUM with other entities of the same type present in the entire dataset, e.g., *2006 model year ⇒ 2008 model year* in Table 1.

    • **in-Instance**: If all the token rationales in evidence $E$ contain two or more PERSON/ORG entities, their positions are randomly *swapped* between different pieces of evidence, e.g., *Ford* (PERSON) ⇔ *Patrick Carpentier* (PERSON) in Table 1.

    • **Consistent Edit**: The same entity token is processed consistently throughout all pieces of evidence, to preserve the multi-hop correlation within the evidence. For example, if an entity is identified in one piece of evidence, it will be consistently replaced or swapped across all pieces of evidence within the instance.

We use the editing rules to produce one edited evidence for each instance based on a random seed. Notably, the PERSON and ORG entities are unique to each instance, rather than across the entire dataset. Thus, we prefer random in-instance swapping over in-dataset replacing to avoid introducing irrelevant information from the dataset into the edited evidence. See examples in Appendix A.

**Ad-Checking** The random operation in our editing rules may raise the under-editing evidence (i.e., $→SUP$) or the over-editing evidence (i.e, $→NEI$) for $SUP$ instances, resulting in the generated claim $c'$ based on this evidence being an incorrect semantic perturbation compared to its original claim $c$. To this end, we use an existing fact verification model to verify the original claim $c$ based on the edited evidence, thus ensuring that this evidence is still valid for further providing to the claim *Generator*. We adopt the RoBERTa (Liu et al., 2019) model, with the concatenation of the edited evidence and the original claim $c$ as input, which is fine-tuned on HOVER (Jiang et al., 2020) dataset with instances labeled as $SUP$, $REF$, and $NEI$. The edited evidence that yields a $REF$ prediction is retained as counterfactual evidence $E'$ (i.e., $(c, E')→REF$). If not, we discard this case for generating counterfactuals. See Appendix B for details.

After Editing and Ad-Checking, we are ready to proceed with the claim generation for the $SUP$ and $REF$ instances. We retain the original $REF$ instance as $(c, E, T, R_s, REF)$, and have perturbed the $SUP$ instance $(c, E, T, R_s, SUP)$ to $(c, E', T', R_s, REF)$, where $T$ and $T'$ denote the set of original and edited causal entities extracted from the token rationales $R_t$, respectively. Up to this step, we generate the counterfactuals $(c, E', REF)$ by altering the causal entities within the multi-hop evidence for $(c, E, SUP)$.

### 3.3 Generator: Claim Generation

As Tan et al. (2023) notes, the direct generation of refuted claims is challenging and may require additional ontology-like mechanisms to ensure that the generation is plausible but reversed. Thus, we opt to generate counterfactual claims $c'$ that are **supported** by the evidence $E/E'$ from the instances. Notably, we do not intervene too much in its generation process, apart from regulating the generated claim $c'$ sensitive to the causal entities $T/T'$. This allows us to ensure the linguistically diverse generation while preserving the factual consistency with evidence $E/E'$.

**Generation** We first use a pre-trained generation model (e.g., T5 (Raffel et al., 2020)) fine-tuned on the $SUP$ instances in FEVEROUS dataset (Aly et al., 2021), using the concatenation of all the gold-standard evidence $E$ as input and the corresponding claim $c$ as the target text (i.e., $E → c$). Unlike prior work on editing the original claim $c$, this encourages the linguistically diverse generation by synthesizing the semantic and correlation information between the multi-hop evidence.

Then, to ensure that the generated claim $c'$ presents factual consistency with the evidence $E/E'$, we apply constrained beam search decoding (Anderson et al., 2017; Post and Vilar, 2018; Hu et al., 2019) with entity constraints to guide the claim generation, by taking the concatenation of all sentence rationales $R_s$ in $E/E'$ as input. Specifically, regarding the list of causal entity tokens $dc_i = [t_{i,1}, t_{i,2}, \ldots, t_{i,j}]$ within each piece

of evidence as disjunctive constraints, where $t_{i,*} \in T/T'$ denotes the causal entity in the $i$-th evidence, and $j$ is the number of the entities, we acquire the conjunction constraint of the beam search by combining of all disjunctive constraints,

$$CONS = dc_1 \wedge dc_2 \wedge \cdots \wedge dc_n, \qquad (1)$$

where $n$ is the number of evidence. The conjunctive constraint during decoding encourages the generated claim $c$ to contain at least one causal entity from each piece of evidence, thus ensuring factual consistency with the multi-hop evidence. After repeated generation, we generate $k$ ($k = 10$ in our experiments) candidate counterfactual claims $C' = \{c'_1, c'_2, ..., c'_k\}$ for each instance.

**Post-Checking**   The claim generation model can be noisy, potentially leading to the non-reversed predictions of a claim $c'$ given $E$. To ascertain the label flipping between claim $c'$ and $c$, i.e, $(c'_i|_{i=1}^{k}, E) \rightarrow y' \neq y$, by taking the concatenation of each candidate counterfactual claim $c'_i$ with its corresponding original evidence $E$ as input, we use the same three-way fact verification model as in Ad-Checking to filter the candidate counterfactual claims. We retain those candidate claims in $C'$ that yield a predicted label $y' \neq y$.

**Discussion**   Claim generation can also be done by very large language models (LLMs) (e.g., ChatGPT (OpenAI, 2022)) with in-context learning (Brown et al., 2020; Wei et al., 2022). However, since our editing may introduce inconsistencies with common sense, we empirically find that the edited evidence $E'$ is more likely to conflict with the internal knowledge of LLMs, thus leading to the irrelevant content or even failure in generating the claim $c'$. Thus, we choose the fine-tuned generation models.

### 3.4   Filtering

Unlike prior work that relies on a curated set of minimal edits (e.g., Yang et al. (2021)), the strategy in our *Generator* maybe over-generate claim $c'$ with over diverse semantic shift compared to $c$. Thus, following Paranjape et al. (2022), we use post-hoc filtering with two modules on generated claims $C'$ to ensure the minimal semantic (Keane and Smyth, 2020) and topic perturbation compared to the original claim $c$.

**Semantic Filtering**   The MoverScore (Zhao et al., 2019), which combines the contextualized representations with the Earth Mover distance (Rubner

et al., 2000), measures the semantic similarity between two sentences. We thus use this metric to calculate *semantic fidelity score* between each counterfactual claim in $C'$ and its corresponding original claim $c$, evaluating the semantic change between these two claims.

**Entity Filtering**   We introduce the *entity fidelity score* by calculating the overlap rate of entities between strings of claim $(c', c)$ pair. This allows us to ensure topic consistency between $c'$ and $c$, filtering out the irrelevant claims from a topic perspective (Si et al., 2021).

One generated claim $c' \in C'$ with the highest sum score over *semantic fidelity score* and *entity fidelity score* is retained for each instance. Finally, our RACE produces the counterfactual data for each instance $(c, E, y)$ in the dataset, including $(c', E, y')$ and $(c, E', y')$.

## 4   Experiments

**Datasets**   We generate counterfactual data for HOVER[2] training set (Jiang et al., 2020), a multi-hop dataset with facts sourced from Wikipedia. We evaluate the model generalization on three types of development sets, (I) In-domain setting (sourced from Wikipedia), including FEVER (Thorne et al., 2018) and FEVEROUS (Aly et al., 2021). (II) Out-of-domain setting (sourced from specific domains), including PolitiHop (political news) (Ostrowski et al., 2021), SCIFACT (scientific articles) (Wadden et al., 2020), HealthVer (Sarrouti et al., 2021) and PubHealth (public health) (Kotonya and Toni, 2020). (III) Challenge setting (contrastive data), including FM2 (Eisenschlos et al., 2021) and VITAMINC (Schuster et al., 2021). Details and statistics of datasets are presented in Appendix C.

**Baselines**   We use three types of baselines to augment the HOVER training set, (I) Data augmentation method: EDA (Wei and Zou, 2019). (II) Counterfactual data augmentation methods: CrossAug (Lee et al., 2021) and POLYJUICE (Wu et al., 2021). (III) LLMs: GPT-3 (text-davinci-003) (Brown et al., 2020) and ChatGPT (gpt-3.5-turbo-0301) (OpenAI, 2022). More details are presented in Appendix D.

---

[2]Since the HOVER dataset contains explicit multi-hop correlation among evidence based on different reasoning type, we choose it to generate counterfactuals and report results in this paper.

Table 2:

| Source of data | $|D_{train}|$ | In-domain | | | Out-of-domain | | | | Challenge | |
|---|---|---|---|---|---|---|---|---|---|---|
| | | HOVER | FEVER | FEVEROUS | PolitiHop | SCIFACT | HealthVer | PubHealth | FM2 | VITAMINC |
| None | 18,171 | 82.55 | 76.70 | 69.43 | 48.74 | 62.77 | 54.98 | 53.01 | 61.51 | 67.05 |
| EDA | 36,342 | 82.55 | 73.60 | 68.22 | **54.62** | 62.77 | 53.68 | 45.99 | 60.56 | 59.63 |
| CrossAug | 29,174 | 82.28 | 65.92 | 70.06 | **54.62** | 57.98 | 49.24 | 39.15 | 56.12 | 61.66 |
| POLYJUICE | 25,190 | 81.10 | 76.43 | 67.94 | 45.38 | 57.98 | 54.65 | 46.28 | 57.14 | 62.29 |
| GPT-3 | 24,171 | 80.75 | 72.30 | 67.56 | 51.26 | 64.36 | 49.46 | 42.91 | 61.25 | 62.21 |
| ChatGPT | 24,171 | 80.13 | 77.77 | 69.04 | 44.54 | 60.64 | 51.84 | 45.00 | 50.74 | **68.14** |
| *our* RACE (BART) | 24,398 | 82.78 | 76.07 | 70.63 | 47.06 | 61.17 | 46.97 | 42.81 | 59.17 | 59.54 |
| *our* RACE (GPT-2) | 23,645 | 82.53 | 77.15 | 66.07 | 45.38 | **65.43** | 54.87 | 53.52 | **62.36** | 67.88 |
| *our* RACE (T5-large) | 26,638 | **83.18** | **78.11** | **71.55** | 47.06 | 62.77 | **55.84** | **56.59** | 61.16 | 67.71 |
| *our* RACE (T5-base) | 26,917 | 83.15 | 75.05 | 70.50 | 52.94 | **65.43** | 55.41 | 53.52 | 62.19 | 66.50 |
| -CONS | 28,468 | 82.53 | 73.93 | 70.09 | 48.74 | 59.04 | 52.71 | 49.06 | 62.28 | 67.31 |
| -EDIT | 28,359 | 80.75 | 71.50 | 68.13 | 54.62 | 60.64 | 52.92 | 47.47 | 61.33 | 62.72 |
| -EDIT&CONS | 27,682 | 83.00 | 76.84 | 70.69 | 43.70 | 60.11 | 52.60 | 53.42 | 60.05 | 64.74 |
| $w\,(c, E', REF)$ | | | | | | | | | | |
| *our* RACE (BART) | 27,909 | 83.33 | 76.65 | 69.16 | 41.18 | 59.57 | 54.00 | 44.20 | 61.42 | 66.16 |
| *our* RACE (GPT-2) | 27,156 | 82.78 | 75.31 | 70.52 | 51.26 | 62.77 | 51.62 | 51.83 | 59.97 | 62.18 |
| *our* RACE (T5-large) | 30,149 | 82.90 | 78.69 | 69.29 | 47.90 | 64.89 | 55.41 | 52.03 | 61.08 | 66.31 |
| *our* RACE (T5-base) | 30,428 | 82.63 | 76.73 | 70.90 | 57.14 | 60.11 | 55.63 | 47.87 | 61.33 | 67.66 |

Table 2: Fact verification accuracy of various data augmentation methods on different development sets in three settings. $|D_{train}|$ shows the total number of training instances, including 18,171 original HOVER training instances. $w\,(c, E', REF)$ denotes the incorporation of counterfactual instances $(c, E', REF)$ into the training set. *-CONS* denotes the use of beam search instead of constrained beam search in claim generation. *-EDIT* denotes that the evidence editing stage is skipped, and counterfactual claims are generated directly from the original evidence for each original instance. The best of the main results are marked in bold. The results with further improvement in model performance after the incorporation of $(c, E', REF)$ are boxed.

**Implementation Details** In the experiments, we fine-tune a basic multi-hop fact verification model, an additional RoBERTa-base (Liu et al., 2019), on the original training data $(c, E, y)$ and the counterfactual data generated by each method. The model is evaluated on the development set of different datasets.

For the basic multi-hop fact verification model, we concatenate the claim and all evidence as input sequence, and limit its maximum length to 130. We set the batch size to 4 and optimize the model through a cross entropy loss using the AdamW optimizer (Loshchilov and Hutter, 2019) with the learning rate of 1e-5. For claim generation, we conduct experiments with four generation models: BART-base (Lewis et al., 2020), T5-base, T5-large (Raffel et al., 2020) and GPT-2 (Radford et al., 2019). The beam size is 30 and the max length of generated text is 96.

## 5 Results and Discussion

### 5.1 Main Results

Neglecting the logical relationships within the correlated input results in a failure to generate counterfactual evidence $E'$ for baselines. Thus, we mainly compare the effects of the counterfactual

data $(c', E, y')$ generated by the different methods. Meanwhile, we also report the results after incorporating $(c, E', REF)$ into the training set (bottom of Table 2).

**Out-of-domain Setting** Table 2 shows the effects of the data generated by RACE and baselines on the OOD generalization. We can observe that, (I) RACE significantly improves model performance on PolitiHop, SCIFACT and PubHealth compared to the results without data augmentation, and outperforms baselines on almost all OOD datasets, demonstrating the effectiveness of our augmentation strategy for multi-hop fact verification task. (II) RACE significantly outperforms POLYJUICE, showing that the general-purpose CDA method, designed for tasks without requiring complex reasoning on the input, fails to achieve acceptable results on multi-hop fact verification task, and even impairs the OOD generalization. (III) The counterfactual data generated by LLMs provides little improvement in OOD generalization, demonstrating that CDA for multi-hop fact verification task remains challenging for LLMs by using the in-context learning alone. (IV) The incorporation of $(c, E', REF)$ further improves the model generalization to a certain extent on PolitiHop, indicating

| Original Instance | |
|---|---|
| **Claim** | The 1994 British romantic comedy that Charlotte Ninon Coleman played Scarlett in featured the song "Love". |
| **Evidence** | 1. [Reg Presley] He wrote the song "Love Is All Around", which was featured in the films "Four Weddings and a Funeral" and "Love Actually". 2. [Charlotte Coleman] Charlotte Ninon Coleman (3 April 1968 – 14 November 2001) was an English actress best known for playing Scarlett in the film "Four Weddings and a Funera", Jess in the television drama "Oranges Are Not the Only Fruit", and her childhood roles of Sue in "Worzel Gummidge" and the character Marmalade Atkins. 3. [Four Weddings and a Funeral] Four Weddings and a Funeral is a 1994 British romantic comedy film directed by Mike Newell. |
| **Label** | SUPPORTS |
| Counterfactual Claims | |
| **CrossAug** | The 1994 British romantic comedy that Charlotte Ninon Coleman played Scarlett in featured the song ~~"Love"~~. |
| **POLYJUICE** | The 1994 British romantic comedy that did not win Charlotte Ninon Coleman played Scarlett in featured the song "Love". |
| **ChatGPT** | The 1994 American romantic comedy that Charlotte Ninon Coleman played Scarlett in featured the song "Love". |
| *our* **RACE (T5-base)** | Marmalade Atkins directed the 1948 British romantic comedy that Reg Presley played Charlotte Coleman in. It featured the song "Love". |

Table 3: Examples of counterfactual claims on HOVER training set derived by different methods. The difference between the counterfactual claim and the original claim is highlighted in blue. See Table 5 in Appendix A for the corresponding edited evidence and more examples.

| Method | Flip. ↑ | Flu. ↓ | Sim. ↑ | Div. ↑ | M.h. ↑ |
|---|---|---|---|---|---|
| **CrossAug** | 0.3138 | 209.34 | **0.6100** | 2.24 | **0.6090** |
| **POLYJUICE** | 0.6066 | 195.18 | 0.5969 | 1.20 | 0.5960 |
| GPT-3 | 0.3970 | 96.84 | 0.5873 | 1.47 | 0.5865 |
| **ChatGPT** | 0.4160 | 107.94 | 0.5906 | 1.73 | 0.5898 |
| **RACE (T5-large)** | 0.9402 | **55.03** | 0.5770 | 11.22 | 0.5763 |
| **RACE (T5-base)** | **0.9457** | 55.81 | 0.5770 | 11.19 | 0.5763 |
| *-FILTER* | 0.8388 | 58.83 | 0.5773 | **13.01** | 0.5766 |

Table 4: Automatic intrinsic evaluation results. For **Flip rate (Flip.)**, we use a RoBERTa-based classifier fine-tuned on the HOVER training set to calculate the verification accuracy of the instance $(c', E, y')$. For **Fluency (Flu.)**, following previous work (Atanasova et al., 2020; He et al., 2023), we use the perplexity scores calculated by GPT-2 to evaluate the fluency of $c'$. For **Similarity (Sim.)**, we calculate the MoverScore between $c'$ and $c$. For **Diversity (Div.)**, following Rani et al. (2023), we use the inverse of the BLEU score (Papineni et al., 2002) to measure dissimilarity between $c'$ and $c$. For **Multi hop (M.h.)**, we employ MoverScore to calculate the average semantic similarity between $c'$ and $e_i$, where $e_i \in E$, to evaluate the coherence like He et al. (2023). *-FILTER* denotes the evaluation of all the generated claims before post-checking and filtering stage. The best results are marked in bold.

that the edited evidence still remains multi-hop correlated and reasonable.

**Challenge Setting** Comparing the results on challenging contrastive datasets, as Table 2 shows, training with RACE data improves the fact verification accuracy, while almost all the baselines degrade the performance of the model. This phenomenon confirms that our method improves model robustness to spurious correlations. Additionally, the incorporation of the $(c, E', REF)$ yields no improvement in verification accuracy, probably because these datasets are constructed in response to

the elimination of spurious correlations between features in **claim** and labels.

**In-domain Setting** As shown in Table 2, RACE improves the model performance on in-domain data, while most baselines tend to degrade it. Notably, our method has the most significant improvement on the FEVEROUS development set, which requires four pieces of true evidence to verify each claim on average. This further demonstrates the effectiveness of our method for multi-hop fact verification task.

## 5.2 Ablation Study

We conduct ablation studies on evidence editing and claim generation stage to verify the reasonableness of causal entities in token rationales. All the experiments are conducted based on RACE (T5-base).

Firstly, we use ordinary beam search instead of constrained beam search during the claim generation stage (i.e., *-CONS* in Table 2). The results in Table 2 reveal that a significant performance decrease occurs on both in-domain and OOD data. It might be explained by constraints based on entities in token rationales, which allow the generated claim to be multi-hop and topic consistent with the original claim, resulting in a more efficient counterfactual. In contrast, we note a slight improvement on the challenge datasets, which might be attributed to the shorter length of claims in both datasets (each claim contains about 13 words on average).

Then, we skip the evidence editing stage and directly generate the counterfactual claims for all the instances (i.e, *-EDIT* in Table 2) by a T5-base language model. The model is fine-tuned on FEVER-OUS to generate claims that are supported or re-

futed by the input evidence via setting the prefix. As shown in Table 2, the accuracy decreases substantially on almost all datasets, except for PolitiHop. It can be explained by the fact that political news typically focuses on event information rather than entity information, hence entity-based evidence editing fails to improve model generalization on PolitiHop.

Finally, we further remove both the constrained beam search and evidence editing stage (i.e., -*CONS&EDIT* in Table 2). A significant decrease in accuracy is observed on both OOD and challenge data, which demonstrates that the proposed evidence editing based on rationales and claim generation based on entities are crucial for improving the generalization and robustness of the multi-hop fact verification models.

### 5.3   Intrinsic Evaluation

For further analysis of the quality of the generated counterfactual claims, following Chemmengath et al. (2022) and Dixit et al. (2022), we automatically and manually evaluate the generated counterfactual claims according to the following five criteria: (I) *Flip rate (Flip.)*, measuring if the label of the generated claim is flipped based on the original evidence; (II) *Fluency (Flu.)*, measuring whether the generated claim is grammatically correct and semantically meaningful; (III) *Diversity (Div.)*, reflecting the linguistic diversity of the generated claim compared to the original claim; (IV) *Similarity (Sim.)*, measuring the degree of semantic similarity between the generated claim and the original claim, where we use MoverScore (Zhao et al., 2019) instead of Levenshtein edit distance (Levenshtein et al., 1966) in the automatic evaluation to balance with diversity; (V) *Multi hop (M.h.)*, measuring whether the generated claim is multi-hop and relevant to the evidence.

**Automatic Evaluation**   For a fair comparison, the claims generated before and after the post-checking and filtering are compared with the baselines separately. As shown in Table 4, RACE outperforms baselines significantly in terms of flip rate, diversity, and fluency. It demonstrates the ability of RACE to generate fluent and *linguistically diverse* counterfactual claims based on the edited evidence, while keeping *label flipping* and *logical relationships* with the original evidence. Moreover, the counterfactual claim after the filtering stage achieves a higher flip rate and fluency score com-

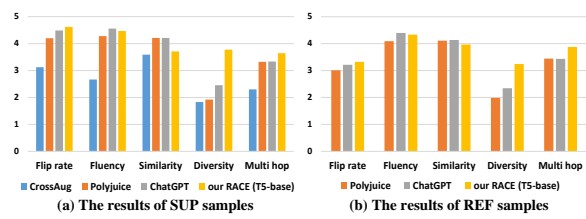

(a) The results of SUP samples   (b) The results of REF samples

Figure 2: The results of human evaluation, where 1 indicates a complete breach of the criteria and 5 indicates full compliance. The inter-rate agreement measured by Krippendorff's $\alpha$ (Krippendorff, 2011) is 0.54.

pared to the one before filtering, which illustrates the necessity of the filtering stage for generating high-quality counterfactual data. For automatic evaluation of *Multi hop*, we follow He et al. (2023) to use MoverScore to evaluate the multi hop of counterfactuals. And all methods achieve comparable results. However, we argue that this is compromised since its solely semantic comparison cannot reflect whether all the evidence can be aggregated as a whole to verify the counterfactual claim.

**Manual Evaluation**   To address the limitations of the automatic evaluation, we adopt the human evaluation to qualify the counterfactuals from different aspects. Specifically, we randomly select 30 $SUP$ instances and 30 $REF$ instances and ask three postgraduate students with an NLP background to score counterfactual claims in a likert scale of 1 to 5 according to the above criteria. Since CrossAug can only generate counterfactuals for $SUP$ instances, we compare the results on $SUP$ and $REF$ instances separately. The evaluation results are shown in Figure 2. It can be observed that RACE well outperforms baselines, particularly in terms of diversity, which illustrates the ability of RACE to generate *human-readable*, *diverse*, and *label-flipping* counterfactual claims. Meanwhile, entity constraint-based generation enables RACE to generate multi-hop claims.

Overall, both the automatic and manual evaluation results show the effectiveness of RACE from different aspects for multi-hop fact verification task.

### 5.4   Qualitative Evaluation

Table 3 presents an example of the original instance and the counterfactual claims generated by different methods. The words that differ from the original claim are highlighted. It can be observed that RACE generates a linguistically diverse and flu-

ent counterfactual claim, and the original label is successfully flipped. Obviously, the counterfactual claim generated by RACE can be combined with the original evidence to form a valid multi-hop fact verification instance, which is logical and can be verified according to the given evidence. Moreover, the claim generated by RACE is semantically and lexically similar to the original claim, benefiting casual entities in multi-hop rationales. Nevertheless, the baselines tend to simply modify the original claim, despite the use of language models. As shown in Table 3, most of the baselines (including LLMs), prefer to add "not" to the original claim or make antonym substitutions. Such modifications make the counterfactual claims lexically similar to the original claim, but are not valid for multi-hop fact verification and cannot generate a diverse and logical counterfactual claim (as evidenced by lower flip rate and diversity in Table 4 and Figure 2).

### 5.5 Effect of Generation Models

We adopt different generation models to test the effect of the generation ability on our method, which aims to illustrate the independence of our proposed method from a particular generation model (i.e., Generation Model-Agnostic). As shown in Table 2, compared to the baselines, our RACE yields a comparable or improved performance based on different generation models, especially the results based on T5-base and T5-large. Besides, We empirically find that different generation models have more prominent performance on specific datasets, e.g., GPT-2 on SCIFACT and FM2 datasets, and T5 on 6 datasets.

To explore the effect of the number of parameters, we further compare the results based on T5-base and T5-large. As Table 4 and 2 shows, compared to T5-base, counterfactuals generated by fine-tuned T5-large are more fluent and linguistically diverse, and further improve the model performance on most datasets. This illustrates that it is possible to further improve the effectiveness of our method by using a more powerful generation model. Thus, for the choice of the generation model, we recommend choosing the powerful possible generation model in the absence of the priors to the data.

### 6 Conclusion

We present a novel rationale-sensitive pipeline counterfactual data augmentation method (RACE) to generate *logical*, *diverse*, and *label-flipping*

counterfactuals for multi-hop fact verification task. An Explain-Edit-Generate architecture is constructed to generate diverse and logical counterfactual claims based on the rationales. Then, a filter process with two modules is employed to further regularize semantic and topic consistency. Experimental results reveal the improvement in OOD generalization and robustness of the proposed method. Intrinsic evaluation and qualitative evaluation of counterfactual claims show that RACE can generate linguistically diverse and label-flipping counterfactual data while preserving logical relationships.

### Limitations

As multi-hop fact verification is a relatively complex reasoning task, designing an effective method to generate counterfactuals for this task requires a consideration of the logical relationships between the claim and the evidence and between multiple pieces of evidence, making our proposed method more complex and cumbersome. Meanwhile, the use of heuristic rules in the editing process results in the inability to generalize to other tasks and the need to recreate the rules. In addition, the prompts given to LLMs for generating counterfactual claims can be further elaborated, e.g., using chain-of-thought, to exploit more potential of LLMs on CDA for multi-hop fact verification task.

In the future, due to the flexible generation of LLMs, we will explore the construction of effective prompts to generate counterfactuals for multi-hop fact verification using the Chain-of-Thought.

### Acknowledgement

The authors would like to thank the anonymous reviewers for their insightful comments. This work is funded by the National Natural Science Foundation of China (62176053) and supported by the Big Data Computing Center of Southeast University. YH is supported by a Turing AI Fellowship funded by the UK Research and Innovation (grant no. EP/V020579/1, EP/V020579/2).

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

## A  Evidence Editing

Table 5 shows examples of the evidence edited by RACE. We can observe that rationale- and entity-based editing enables the edited evidence to still retain multi-hop correlation with each other and present a completely different fact from the original evidence. Hence, the claim generator can generate logical, fluent, and linguistically diverse counterfactual claims based on the edited evidence.

## B  Checking Module

For the ad- and post-checking module, we fine-tune a RoBERTa-base classifier to filter invalid edited evidence and counterfactual claims, respectively. To improve the quality of the retained data, we fine-tune it on the $SUP$, $REF$, and $NEI$ instances rather than just the $SUP$ and $REF$ instances.

Considering that we perform CDA on HoVer training set during the experiment while no $NEI$ instances are available in HoVer, we first conduct data augmentation on HoVer dataset to incorporate $NEI$ instances by perturbing existing instances. Specifically, for a random instance in HoVer, we randomly remove one piece of true evidence or randomly pair the claim with the evidence of another instance. To avoid imbalance classes, we randomly select half of the $SUP$ instances and half of the $REF$ instances for perturbation and each perturbation strategy is employed with equal probability. Finally, the fine-tuned RoBERTa-base classifier has 81.23% on label accuracy of claim verification on $NEI$ augmented HoVer development set. The statistics of $NEI$ augmented HoVer are shown in Table 6.

Other implementation details are the same as the fact verification model in the OOD generalization experiment described in Section 4.

## C  Datasets

- **HoVer** (Jiang et al., 2020), a dataset for multi-hop fact verification, which challenges models to extract relevant evidence from several Wikipedia articles and verify whether the claim is SUPPORTED or REFUTED by the evidence. We construct the dataset following Khattab et al. (2021), where each claim is associated with five pieces of evidence.

- **FEVER** (Thorne et al., 2018), a large-scale fact verification dataset with the claims generated by altering sentences extracted from Wikipedia. The claims in FEVER are classified as SUPPORTS, REFUTES or NOT ENOUGH INFO (NEI) by annotators and more than 87% of them only require information from a single Wikipedia article. We remove the instances with NEI label and only retain the other two classes of instances in our experiments.

- **FEVEROUS** (Aly et al., 2021), a large-scale multi-hop fact verification dataset consisting of claims verified against Wikipedia pages and labeled as SUPPORTS, REFUTES or NOT ENOUGH INFO (NEI). Each claim has evidence in the form of sentences and/or cells from tables on Wikipedia. Following Chen et al. (2020) and Si et al. (2023b), we employ the simple table linearization template to generate contextualized sequence representations for table evidence. We remove the instances with NEI label and only retain the other two classes of instances in our experiments.

- **PolitiHop** (Ostrowski et al., 2021), a multi-hop fact verification dataset of real-world claims with manual annotations of evidence from PolitiFact articles. The labels include FALSE, HALF-TRUE and TRUE. In our experiments, we remove the instances with HALF-TRUE label and only retain the other two classes of instances.

- **SciFact** (Wadden et al., 2020), a scientific fact verification dataset of 1.4K expert-written scientific claims paired with evidence. As with the above dataset, we only retain the instances with SUPPORTS and REFUTES labels to evaluate the model.

| Original Instance | | |
|---|---|---|
| **Claim** | The 1994 British romantic comedy that Charlotte Ninon Coleman played Scarlett in featured the song "Love". | |
| **Evidence** | 1. [Reg Presley] He wrote the song "Love Is All Around", which was featured in the films "Four Weddings and a Funeral" and "Love Actually". 2. [Charlotte Coleman] Charlotte Ninon Coleman (3 April 1968 – 14 November 2001) was an English actress best known for playing Scarlett in the film "Four Weddings and a Funera", Jess in the television drama "Oranges Are Not the Only Fruit", and her childhood roles of Sue in "Worzel Gummidge" and the character Marmalade Atkins. 3. [Four Weddings and a Funeral] Four Weddings and a Funeral is a 1994 British romantic comedy film directed by Mike Newell. | |
| **Label** | SUPPORTS | |

| Edited Evidence | | |
|---|---|---|
| **Edired Evidence** | 1. [Mike Newell] He wrote the song "Love Is All Around", which was featured in the films "Four Weddings and a Funeral" and "Love Actually". 2. [Reg Presley] Reg Presley (3 August 1987 – 26 June 2000) was an English actress best known for playing Charlotte Coleman in the film "Four Weddings and a Funera", Scarlett in the television drama "Oranges Are Not the Only Fruit", and her childhood roles of Sue in "Worzel Gummidge" and the character Jess. 3. [Four Weddings and a Funeral] Four Weddings and a Funeral is a 1948 British romantic comedy film directed by Marmalade Atkins. | |

| Counterfactual Claims | | |
|---|---|---|
| **CrossAug** | The 1994 British romantic comedy that Charlotte Ninon Coleman played Scarlett in featured the song ~~"Love"~~. |
| **POLYJUICE** | The 1994 British romantic comedy that did not win Charlotte Ninon Coleman played Scarlett in featured the song "Love". |
| **ChatGPT** | The 1994 American romantic comedy that Charlotte Ninon Coleman played Scarlett in featured the song "Love". |
| *our* **RACE** | Marmalade Atkins directed the 1948 British romantic comedy that Reg Presley played Charlotte Coleman in. It featured the song "Love". |

| Original Instance | | |
|---|---|---|
| **Claim** | Bruce Geller who died in 1978 developed American television detective show Mannix. | |
| **Evidence** | 1. [Mannix] Created by Richard Levinson and William Link and developed by executive producer Bruce Geller, the title character, Joe Mannix, is a private investigator. 2. [Bruce Geller] Bruce Bernard Geller (October 13, 1930 – May 21, 1978) was an American lyricist, screenwriter, director, and television producer. | |
| **Label** | SUPPORTS | |

| Edited Evidence | | |
|---|---|---|
| **Edired Evidence** | 1. [Mannix] Created by Joe Mannix and Richard Levinson and developed by executive producer William Link, the title character, Bruce Geller, is a private investigator. 2. [William Link] William Link (December 14, 1898 – April 30, 1977) was an American lyricist, screenwriter, director, and television producer. | |

| Counterfactual Claims | | |
|---|---|---|
| **CrossAug** | Bruce Geller who passed away in 1978 developed American television detective show Mannix. |
| **POLYJUICE** | Bruce Geller who died in 1978, did not developed American television detective show Mannix. |
| **ChatGPT** | Bruce Geller, who passed away in 1985, developed the American television detective show Mannix. |
| *our* **RACE** | The executive producer of American television detective show Mannix died in 1877. The show was created by Joe Mannix and Richard Levinson. |

Table 5: Examples of edited evidence and counterfactual claims on HOVER training set. The differences from the original instance are highlighted in blue.

| Augmented HOVER | Num.SUP | Num.REF | Num.NEI | Total |
|---|---|---|---|---|
| **Train** | 11,023 | 7,148 | 9,086 | 27,572 |
| **Dev** | 2,000 | 2,000 | 2,000 | 6,000 |

Table 6: The statistics of augmented HOVER with $NEI$ instances. Num.SUP, Num.REF and Num.NEI are the number of $SUP$ instances, $REF$ instances, and $NEI$ instances, respectively.

- **HealthVer** (Sarrouti et al., 2021), an evidence-based fact verification dataset for health-related claims, where the relations between each piece of evidence and the associated claim are manually annotated as SUPPORT, REFUTE, and NEUTRAL. We remove the instances with the NEUTRAL label. As the evidence provided by HealthVer contains several sentences, we split it into multiple pieces of evidence to simulate a multi-hop scenario.

- **PubHealth** (Kotonya and Toni, 2020), a 4-way classification dataset for explainable fact verification with gold standard explanations by journalists in the public health setting. We only retain the instances with TRUE and FALSE labels, and the explanation provided is split into separate sentences as multiple pieces of evidence.

Given an original claim with corresponding evidence and label (SUPPORTS or REFUTES), generate a counterfactual claim based on the evidence, taking care to ensure that the generated counterfactual claim is as **similar** as possible to the original claim, while being aware of linguistic **diversity** and the **change** of labels.

**Example:**

**Claim:** Bettany Hughes, an English historian scholar, born May 15th, 1967, presented "The Spartans".

**Evidence:**

The Spartans (documentary): " The Spartans " was a 3-part historical documentary series first broadcast on UK terrestrial Channel 4 in 2003, presented by Bettany Hughes.

Bettany Hughes: Bettany Hughes ( born May 15 , 1967 ) is an English historian, author, and broadcaster.

**Label:** SUPPORTS

**Generate a counterfactual claim:**

"The Spartans" is a documentary presented by Bettany Hughes, an American historian scholar born on March 24, 1980.

**Claim:** The writer Norman Alfred William Lindsay enjoyed boxing, but the author of The Hundred Secret Senses did not.

**Evidence:**

Amy Tan: Amy Tan ( born February 19, 1952 ) is an American writer whose works explore mother-daughter relationships and the Chinese American experience.

The Hundred Secret Senses: The Hundred Secret Senses is a bestselling 1995 novel by Chinese-American writer Amy Tan.

Norman Lindsay: Norman Alfred William Lindsay ( 22 February 1879 – 21 November 1969 ) was an Australian artist, etcher, sculptor, writer, editorial cartoonist, scale modeller, and an accomplished amateur boxer.

**Label:** SUPPORTS

**Generate a counterfactual claim:**

Table 7: An example of prompt given to GPT-3 and ChatGPT for generating counterfactual claims.

| Dataset | Num.SUP | Num.REF | Total |
|---|---|---|---|
| HOVER Dev | 2,000 | 2,000 | 4,000 |
| FEVER Dev | 6,666 | 6,666 | 13,332 |
| FEVEROUS Dev | 3,908 | 3,481 | 7,389 |
| PolitiHop Dev | 21 | 98 | 119 |
| SCIFACT Dev | 124 | 64 | 188 |
| HealthVer Dev | 533 | 391 | 924 |
| PubHealth Dev | 628 | 544 | 1,172 |
| FM2 Dev | 596 | 573 | 1,169 |
| VITAMINC Dev | 31,484 | 22,528 | 54,012 |

Table 8: The statistics of the datasets we used in our experiments.

- **FM2** (Eisenschlos et al., 2021), a large-scale dataset of challenging claim-evidence pairs collected through a fun multi-player game which encourages adversarial instances and drastically lowers the number of the instances with "shortcuts". All the claims need to be verified $\in$ {SUPPORTS, REFUTES}.

- **VITAMINC** (Schuster et al., 2021), a large-scale contrastive fact verification dataset, where each contrastive claim is manually written by annotators based on Wikipedia revisions. We only retain the instances with SUP-

PORTS and REFUTES labels in our experiments.

We only test the performance of the basic multi-hop fact verification model on the development set of the above datasets in our experiments. The statistics are shown in Table 8.

## D  Baselines

In our experiments, we compare our method with the following baselines.

- **EDA** (Wei and Zou, 2019), a data augmentation method that applies four simple operations, including synonym replacement, random insertion, random swap, and random deletion, to original sentences to generate new instances.

- **CrossAug** (Lee et al., 2021), a counterfactual data augmentation method that employs a two-stage augmentation pipeline to generate contrastive claims and evidence from existing $SUP$ instances.

- **POLYJUICE** (Wu et al., 2021)), a general-purpose counterfactual generator based on fine-tuned GPT-2 that allows for control over perturbation types and locations.

- **GPT-3** (`text-davinci-003`) ([Brown et al., 2020](#)), a large autoregressive language model with superb few-shot and in-context learning capabilities.

- **ChatGPT** (`gpt-3.5-turbo-0301`) ([OpenAI, 2022](#)), a powerful GPT-3 based model which is trained to follow an instruction in a prompt and provide a detailed response.

For EDA[3] and CrossAug[4], all the experimental setups of them are followed from the original papers and all hyperparameters are set to the same values as in the official code. For POLYJUICE[5], we set the control code to "negation", the beam size to 10, and generate one counterfactual claim for each original claim. All the inputs to the above baselines are only the original claim.

For GPT-3 and ChatGPT, we make use of the APIs provided by OpenAI[6] for generating counterfactual claims and design a prompt with a task introduction and demonstration as input, as shown in the Table 7.

---

[3] https://github.com/jasonwei20/eda_nlp
[4] https://github.com/minwhoo/CrossAug
[5] https://github.com/tongshuangwu/polyjuice
[6] https://openai.com/product