# OpenReview forum: "EXPLAIN, EDIT, GENERATE: Rationale-Sensitive Counterfactual Data Augmentation for Multi-hop Fact Verification"
_EMNLP/2023/Conference — EMNLP 2023 Main_

### Official Review · Reviewer_gZJT · 2023-08-01

**Soundness:** 3

**Excitement:**

3: Ambivalent: It has merits (e.g., it reports state-of-the-art results, the idea is nice), but there are key weaknesses (e.g., it describes incremental work), and it can significantly benefit from another round of revision. However, I won't object to accepting it if my co-reviewers champion it.

**Missing References:**

N/A

**Paper Topic And Main Contributions:**

The main focus of this paper is the Counterfactual Data Augmentation (CDA) method for multi-hop fact verification tasks. The paper argues that current CDA methods mainly cater to single-sentence tasks and struggle to handle the complex logical relations in multi-hop fact verification tasks. To address this issue, a new pipeline method called RACE is proposed, which can generate logically coherent, linguistically diverse, and label-flipping counterfactual data to enhance the generalization and robustness of multi-hop fact verification models.

The main contributions of this paper include:

1. The proposal of an Explain-Edit-Generate framework, which generates counterfactual evidence that is logically consistent with the original evidence but differs factually, based on the extracted rationales for evidence editing and statement generation.

2. During the statement generation, a constrained beam search and entity constraint are utilized to ensure the generated counterfactual statements are semantically diverse and have multi-hop relations with the evidence.

3. The introduction of a Checking and Filtering module, which regularizes the generated counterfactual data from aspects of semantic reasonability and topic consistency.

4. Experimental results demonstrate that the counterfactual data generated using the method in this paper can improve the model performance on in-domain, out-of-domain, and adversarial datasets. At the same time, the counterfactual statements generated in this study exhibit superior logical coherence and linguistic diversity.

**Questions For The Authors:**

N/A

**Reasons To Accept:**


Based on my understanding of this paper, it may bring the following benefits:

1. It provides a new approach for enhancing the generalization and robustness of models for multi-hop fact verification tasks. While counterfactual data augmentation is gaining traction in NLP, research focused on complex reasoning tasks is relatively scarce. This paper provides valuable exploration into how to generate high-quality counterfactual data for such tasks.

2. It designs a targeted counterfactual data generation pipeline, considering the characteristics of multi-hop verification tasks, and generating high-quality training data in terms of logic and semantics. This could serve as a reference or foundation for future research.

3. It introduces the concept of rationales extracted using interpretability methods, focusing on the causality characteristics of the text. This attempt to integrate machine learning interpretability can provide certain inspirations and references for related research.

4. It enriches the application of counterfactual learning in the NLP field, demonstrating the potential of this technique in enhancing model generalization capabilities. This can promote more research focusing on data augmentation and robustness.

5. It empirically verifies that counterfactual data augmentation has a good effect in improving the generalization ability of multi-hop fact verification. This provides new insights for research in this task.

**Reasons To Reject:**

The potential shortcomings and assumed risks in the paper could be:

1. The proposed method has multiple pipeline stages, which are quite complex and might be difficult to adapt to other tasks or datasets. Dependence on heuristic rules in the editing process also reduces generalization capabilities.

2. Although the results show improvement, it is still limited. For instance, the advantage is not significant in some out-of-domain results. More analysis is needed on its robustness and generalization.

3. There could be a more thorough exploration of using large language models for generation. The prompts given to them can be improved to fully leverage their potential.

4. Apart from manually analyzing some samples, there is no manual evaluation of the counterfactual statements. The existing sample analysis is not sufficient and more rigorous manual evaluations should be undertaken.

**Reproducibility:**

4: Could mostly reproduce the results, but there may be some variation because of sample variance or minor variations in their interpretation of the protocol or method.

**Reviewer Confidence:**

3: Pretty sure, but there's a chance I missed something. Although I have a good feel for this area in general, I did not carefully check the paper's details, e.g., the math, experimental design, or novelty.

**Typos Grammar Style And Presentation Improvements:**

While reading the paper in detail, I noticed some issues with language expression that need to be improved:

1. On line 07, I suggest changing "minimally altering" to "minimal perturbations" for a more precise expression.
2. On line 459, change "provides" to "provide" to ensure subject-verb agreement.
3. On line 126, change "reveals" to "reveal" for subject-verb agreement.

Furthermore, regarding the structure of the paper, I suggest:

The section on experimental results could further enhance result analysis, such as discussing more about the reasons behind the results, extracting main findings, etc., to enrich the content of this section.

---

> ### Author Rebuttal · Authors · 2023-08-29
>
> Thanks for your constructive comments and suggestions, and they are exceedingly helpful to improve our paper. We will re-proofread the paper and revise the typo grammar style and presentation based on your suggestions. Furthermore, our point-to-point responses to your comments are given below.
>
> >**Q1**:
> The proposed method has multiple pipeline stages, which are quite complex and might be difficult to adapt to other tasks or datasets. Dependence on heuristic rules in the editing process also reduces generalization capabilities.
>
> **R1**:
>
> Due to the complexity of the multi-hop fact verification task, to generate valid and effective counterfactuals for this task, we design a relative complex pipeline method by regularizing the different properties (i.e., logical relationship, label flipping, and diversity) at different stage during our generation process. In fact, we empirically find that it is very challenging to generate valid counterfactuals using an end-to-end method, even with the LLMs (e.g., *ChatGPT* in Tab.2).
>
> To make our method reproducible, we have simplified the heuristic rules as much as possible by only considering the **entity type** in the editing process. Even though, to ensure the effectiveness of the proposed method on multi-hop fact verification task, we have unfortunate to sacrifice the generalization capabilities of the method on other tasks.
>
> >**Q2**:
> Although the results show improvement, it is still limited. For instance, the advantage is not significant in some out-of-domain results. More analysis is needed on its robustness and generalization.
>
> **R2**:
>
> In fact, substantial generalization improvement is difficult due to the complexity of the multi-hop fact verification task. Compared to the original model (i.e., **None** in Tab.2), we empirically find that the previous methods tend to decrease the generalization ability of the model in most datasets, e.g., in Tab.2, the *CrossAug* shows a decrease in accuracy of 5.74% and 13.86% on HealthVer and PubHealth, respectively. However, our proposed method results in a comparable or better performance on the generalization of the model in most cases, illustrating the effectiveness of our proposed method. We will report the standard deviation of each method to validate the robustness of our method.
>
> >**Q3**:
> There could be a more thorough exploration of using large language models for generation. The prompts given to them can be improved to fully leverage their potential.
>
> **R3**:
>
> We have experimentally found that it is difficult for LLMs to generate counterfactuals with one step for multi-hop fact verification tasks via in-context learning. So far, we have attempted a variety of complex prompts for LLMs, but still failed to achieve satisfactory results. Please refer to the Response **R3** to **Reviewer 1** for details. A potential way is to introduce complex prompts such as Chain-of-Thought, Graph-of-Thought, Tree-of-Thought, etc., so that LLMs can accomplish the counterfactuals generation step by step. That's what we'll explore in the future.
>
> >**Q4**:
> Apart from manually analyzing some samples, there is no manual evaluation of the counterfactual statements. The existing sample analysis is not sufficient and more rigorous manual evaluations should be undertaken.
>
> **R4**:
>
> Indeed, we conduct a thorough manual evaluation of the generated counterfactual claims as Section 5.3 described. The results are shown in Fig.2. Note that we only manually evaluate the counterfactual claims since the previous method cannot generate the counterfactual evidence as our proposed method. The description in line 564 of the paper seems to be a bit off, and we will clarify this point in the revised paper.
>
>
> >**Q5**:
> The section on experimental results could further enhance result analysis, such as discussing more about the reasons behind the results, extracting main findings, etc., to enrich the content of this section.
>
> **R5**:
>
> Thank you for your constructive suggestions, we will report more results and the corresponding analysis to enhance the content of our experiment in the revised paper.

---

### Official Review · Reviewer_HUk1 · 2023-08-05

**Soundness:** 4

**Excitement:**

4: Strong: This paper deepens the understanding of some phenomenon or lowers the barriers to an existing research direction.

**Paper Topic And Main Contributions:**

This paper proposes a counterfactual data augmentation method to handle the multi-hop fact verification task. An Explain-Edit-Generate architecture is leveraged to generate diverse and fluent counterfactuals, and the checking and filtering modules are proposed to regularize the counterfactual data. Experiments demonstrate the effectiveness of the proposed method.

**Reasons To Accept:**

- The method is explained clearly and easy to follow.
- The proposed method has high interpretability.
- Introducing the idea of counterfactual data augmentation seems novel.
- Experiments are solid and comprehensive.

**Reasons To Reject:**

- This paper adopts three generation models as the backbone model and compare their performances with SOTA methods such as GPT-3 and ChatGPT. In my opinion, GPT-3/ChatGPT has far more parameters than BART/GPT-2/T5, whether such comparison is fair? Can the proposed method adapts well to those huger LLM models?

**Reproducibility:**

3: Could reproduce the results with some difficulty. The settings of parameters are underspecified or subjectively determined; the training/evaluation data are not widely available.

**Reviewer Confidence:**

2: Willing to defend my evaluation, but it is fairly likely that I missed some details, didn't understand some central points, or can't be sure about the novelty of the work.

---

> ### Author Rebuttal · Authors · 2023-08-29
>
> Thanks for your encouraging words and constructive comments. In the following, your comments are first stated and then followed by our point-by-point responses.
>
> >**Q1**:
> This paper adopts three generation models as the backbone model and compare their performances with SOTA methods such as GPT-3 and ChatGPT. In my opinion, GPT-3/ChatGPT has far more parameters than BART/GPT-2/T5, whether such comparison is fair? Can the proposed method adapts well to those huger LLM models?
>
> **R1**:
>
> As your mentioned, our method generates counterfactual claims by fine-tuning generation models with relative less parameters, instead of using in-context learning to bootstrap LLMs with frozen parameters. Although GPT-3/ChatGPT has far more parameters than BART/GPT-2/T5, the performance of our method still outperforms these LLMs, which validates the effectiveness of our proposed method.
>
> To explore the effect of the number of parameters, we conducted additional experiments on the *T5-large*. We empirically find that compared to T5-base, counterfactuals generated by fine-tuned T5-large are more fluent and linguistically diverse, and further improve the model performance on most datasets. This illustrates that it is possible to further improve the effectiveness of our method by using a more powerful generation model. The results are shown in the tables below.
>
> During our experiments, we empirically find that it is difficult for LLMs to generate high quality of counterfactuals according to our framework, and no improved performance can be observed. We will release the source code to promote the further research in this filed.
>
> | Source of CFs   | HoVer | FEVER | FEVEROUS | PolitiHop |SCIFACT | HealthVer | PubHealth | FM2 | VitaminC |
> |:----------------|:-----:|:---------:|:---------:|:--------:|:---------:|:--------:|:---------:|:---------:|:--------:|
> | RACE (T5-base)  | 83.15 | 75.05     | 70.50     | 52.94    | 65.43    | 55.41   | 53.52    | 62.19     | 66.50    |
> | RACE (T5-large) | **83.18** | **78.11**   | **71.55**     | 47.06   | 62.77    | **55.84** |**56.59**    | 61.16    | **67.71**    |
>
> Table 1: Fact verification accuracies of data augmentation methods on different development set.
>
> | Method | Flip.$\uparrow$ | Flu.$\downarrow$ | Sim.$\uparrow$ | Div.$\uparrow$ | M.h.$\uparrow$|
> |:------|:-----:|:---------:|:---------:|:--------:|:------:|
> | RACE (T5-base)  | 0.9457 | 55.81 | 0.5770 | 11.19 | 0.5763 |
> | RACE (T5-large) | 0.9402 | **55.03** | 0.5770 | **11.22** | 0.5763 |
>
> Table 2: Automatic intrinsic evaluation results.

---

### Official Review · Reviewer_gQhw · 2023-08-05

**Soundness:** 3

**Excitement:**

4: Strong: This paper deepens the understanding of some phenomenon or lowers the barriers to an existing research direction.

**Paper Topic And Main Contributions:**

The authors describe a method to generate counterfactual data for a multi-hop fact verification task, specifically by passing original instances through RACE, an Explain-Edit-Generate pipeline. First, the Explainer extracts sentence and token rationales from the given multi-hop evidence. Then, the Editor identifies certain named entities among the token rationales, and either replaces or permutes them, making sure that the edited evidence now refutes the original claim. The Generator then creates a counterfactual claim supported by the edited evidence (and refuted by the original evidence). After filtering the counterfactual claims for minimal semantic shift from the original claim, the authors train a basic multi-hop fact verification model on the augmented dataset, comparing it to models trained on the original data as well as on datasets augmented with (counterfactual) instances from other sources. They find that their counterfactual data augmentation scheme results in comparable or better results on the multi-hop fact verification task, while the generated counterfactuals outscore those from other sources in flip rate, fluency, and diversity.

**Reasons To Accept:**

This paper describes some very interesting ideas regarding how to generate counterfactuals for multi-hop fact verification, that are worthy of further exploration. The Explain-Edit-Generate pipeline intuitively makes a lot of sense given the goals. The authors successfully show that their generated counterfactuals are linguistically diverse and label flipping.

**Reasons To Reject:**

The authors state that previous methods of generating counterfactuals, based on making local edits to the claim, struggle to preserve “logical relationships” between multiple hops of evidence. Furthermore, they state that the edits made by the baselines tested are “not valid for multi-hop fact verification” (lines 603-604). No evidence is given for either of these claims, and in fact, Table 4 shows that the baselines achieve a higher “Multi hop” score than their method.

They also claim that their approach “outperforms the SOTA baselines” (line 24). However, the results in Table 2 seem more mixed to me. A RACE model achieves the highest accuracy on six of the nine datasets, but the results vary widely based on the generation model used (BART vs. GPT-2 vs. T5). Also, the authors do not mention how many runs were performed; given the small effect sizes (in most cases, the difference between the best RACE model and the best baseline is within one or two points), I am unsure how robust the results are.

When using GPT-3 and ChatGPT to generate counterfactuals, it seems that those models are given an example of a counterfactual claim in the prompt (as in Table 7). My concern is that it may be possible to engineer GPT-3/ChatGPT to generate counterfactuals with certain properties, based on the example counterfactual. For instance, if in the prompt, the example counterfactual is lexically similar to the original claim, then the counterfactuals generated by GPT-3/ChatGPT may analogously be lexically similar to the given claim, i.e., score lower on diversity. More detail should be given about how the prompt was designed.

**Reproducibility:**

4: Could mostly reproduce the results, but there may be some variation because of sample variance or minor variations in their interpretation of the protocol or method.

**Reviewer Confidence:**

3: Pretty sure, but there's a chance I missed something. Although I have a good feel for this area in general, I did not carefully check the paper's details, e.g., the math, experimental design, or novelty.

**Typos Grammar Style And Presentation Improvements:**

The paper would benefit from another round of proofreading. For example:

line 3: delete “with”

line 4: “brutal” → I would just say “poorly” or something like that

line 7: “is” → “are”

figure 2: “inner-rate” → “inter-rater”

etc.

Also, while the paper usually uses the term “logical” to refer to the logical relationships between multiple hops of evidence, in line 605 (and possibly line 131?) it is used to mean “label flipping”; this should be made consistent.

---

> ### Author Rebuttal · Authors · 2023-08-29
>
> Thank you for reviewing our paper and for the insightful comments, and they are helpful for us to improve our work. We will re-proofread the paper and revise the typo grammar style and presentation based on your suggestions. Furthermore, our point-to-point responses to your comments are given below.
>
> >**Q1**:
> The authors state that previous methods of generating counterfactuals, based on making local edits to the claim, struggle to preserve “logical relationships” between multiple hops of evidence. Furthermore, they state that the edits made by the baselines tested are “not valid for multi-hop fact verification” (lines 603-604). No evidence is given for either of these claims, and in fact, Table 4 shows that the baselines achieve a higher “Multi hop” score than their method.
>
> **R1**:
>
> Multi-hop fact verification is a task that necessitates **the inference of logical relationships between claim and evidence, as well as within the multiple pieces of evidence**, i.e., by aggregating the multi-hop evidence to verify the truthfulness of the claim. As shown in the third row of Tab.1, there are complex relationships between claim and multi-hop evidence, and within the multiple pieces of evidence.
>
> **(I) Qualitative Description**: Existing CAD methods generate counterfactual claims solely by modifying the original claim (e.g., *CrossAug* deletes the words in the claim), rather than using the information of multi-hop evidence. We empirically find that this way will **compromise the semantic integrity of the generated counterfactuals**. As the *CrossAug* shown in Tab.3, the counterfactual claim is generated by removing the entity "Love" (the name of the song), while this entity is the key word to connect the claim and evidence E1, as well as the evidence {E1,E2,E3}. Without this information, we are unsure that whether the same song that the claim and evidence discussed, which results in the multiple pieces of evidence cannot be aggregated as a  whole to verify the claim, thus producing the invalid counterfactuals.
>
> **(II) Quantitative Evaluation**: To our knowledge, there currently exists no automatic metrics to evaluate the logical relationship within the instance. In our paper, we adopt the metric *MoverScore* to evaluate the *Multi hop* of the counterfactuals, while this is compromised. The more reasonable approach to evaluate the logical relationship is adopting the **Human Evaluation**. To this end, we ask three postgraduate students to evaluate the generated counterfactuals of different methods, as shown in Fig.2, our RACE outperforms all other baselines on *Multi hop*, which validate the effectiveness of our method.
>
> Furthermore, *label flipping* is a basic factor in the validity of the counterfactual instances. The lower *label flipping rates* of baselines (as shown in Tab.4 and Fig.2) also present the invalidity of counterfactuals generation by previous methods.
>
> >**Q2**:
> They also claim that their approach “outperforms the SOTA baselines” (line 24). However, the results in Table 2 seem more mixed to me. A RACE model achieves the highest accuracy on six of the nine datasets, but the results vary widely based on the generation model used (BART vs. GPT-2 vs. T5). Also, the authors do not mention how many runs were performed; given the small effect sizes (in most cases, the difference between the best RACE model and the best baseline is within one or two points), I am unsure how robust the results are.
>
> **R2**:
>
> **(I)** We adopt different generation models to test the effect of the generation ability on our method. Overall, compared to the baselines, our method yields a comparable or improved performance based on different generation model, especially the results based on T5-base. We empirically find that different generation models have more prominent performance on specific datasets, e.g., *BART* on *FEVEROUS* dataset, *T5* on 5 datasets. In fact, by experiments conducted on the *T5-large*, we find that the performance can be further improved on most datasets, as shown in the table below. We will report more experimental results and the detailed analysis in the revised paper.
>
> **(II)** The results reported in Tab.2 are mean of the accuracy over three different seeds, we will report the standard deviation of all the methods in the revised paper to illustrate the robust of our method.
>
> | Source of CFs   | HoVer | FEVER | FEVEROUS | PolitiHop |SCIFACT | HealthVer | PubHealth | FM2 | VitaminC |
> |:----------------|:-----:|:---------:|:---------:|:--------:|:---------:|:--------:|:---------:|:---------:|:--------:|
> | RACE (T5-base)  | 83.15 | 75.05     | 70.50     | 52.94    | 65.43    | 55.41   | 53.52    | 62.19     | 66.50    |
> | RACE (T5-large) | **83.18** | **78.11**   | **71.55**     | 47.06   | 62.77    | **55.84** |**56.59**    | 61.16    | **67.71**    |
>
>
>
>
> >**Q3**:
> When using GPT-3 and ChatGPT to generate counterfactuals, it seems that those models are given an example of a counterfactual claim in the prompt (as in Table 7). My concern is that it may be possible to engineer GPT-3/ChatGPT to generate counterfactuals with certain properties, based on the example counterfactual. For instance, if in the prompt, the example counterfactual is lexically similar to the original claim, then the counterfactuals generated by GPT-3/ChatGPT may analogously be lexically similar to the given claim, i.e., score lower on diversity. More detail should be given about how the prompt was designed.
>
> **R3**:
>
> Through extensive experiments on GPT-3/ChatGPT, we have found that such a simple prompt performs better than the other prompts.
>
> **(I)** We attempt the examples with linguistically diverse and lexically distinct from the original claim. However, it has found that LLMs tend to improve the linguistic diversity of counterfactuals by replacing the keywords with irrelevant entities during generation (e.g., *Charlotte Ninon Coleman* in Tab.3 --> *Charlotte Bronte*), resulting in counterfactual claims can not be verified.
>
> **(II)** We attempt the examples with various reasoning types. Unfortunately, we experimentally found that LLMs have difficulty in understanding the multi-hop reasoning type, resulting in the poor quality of generated counterfactual claims. Furthermore, the generalization of the models trained on these counterfactuals declined on the vast majority of out-of-domain datasets.
>
> The following is a comparison of the prompt and the corresponding generated counterfactual claim. We can observe that the Prompt with various reasoning types generates worse results compared to our simple prompt, i.e., LLMs cannot generate counterfactual claims well around the anchor entity **boxing**.
>
> ***Prompt in our paper:***
> ```
> Given a original claim with corresponding evidence and label (SUPPORTS or REFUTES), generate a counterfactual claim based on the evidence, taking care to ensure that the generated counterfactual claim is as similar as possible to the original claim, while being aware of linguistic diversity and the change of labels.
> Example:
> Claim: Bettany Hughes, an English historian scholar, born May 15th, 1967, presented "The Spartans".
> Evidence:
> The Spartans (documentary): “ The Spartans ” was a 3-part historical documentary series first broadcast on UK terrestrial Channel 4in 2003 , presented by Bettany Hughes .
> Bettany Hughes: Bettany Hughes ( born May 15 , 1967 ) is an English historian , author , and broadcaster .
> Label: SUPPORTS
> Generate a counterfactual claim:
> “The Spartans” is a documentary presented by Bettany Hughes, an American historian scholar born on March 24, 1980.
>
> Claim: The writer Norman Alfred William Lindsay enjoyed boxing, but the author of The Hundred Secret Senses did not.
> Evidence:
> Amy Tan: Amy Tan ( born February 19 , 1952 ) is an American writer whose works explore mother-daughter relationships and the Chinese American experience .
> The Hundred Secret Senses: The Hundred Secret Senses is a bestselling 1995 novel by Chinese-American writer Amy Tan .
> Norman Lindsay: Norman Alfred William Lindsay ( 22 February 1879 – 21 November 1969 ) was an Australian artist , etcher , sculptor , writer , editorial cartoonist , scale modeller , and an accomplished amateur boxer .
> Label: SUPPORTS
> Generate a counterfactual claim:
> ```
> ***Generated counterfactual claim:***
> ```
> The writer Norman Alfred William Lindsay not only enjoyed boxing, but he also authored The Hundred Secret Senses.
> ```
>
> ***Prompt with examples of various reasoning types: (E denotes evidence and C denotes claim.)***
> ```
> Given a original claim with corresponding evidence and label (SUPPORTS or REFUTES), generate a counterfactual claim based on the evidence, taking care to ensure that the generated counterfactual claim is as similar as possible to the original claim, while being aware of linguistic diversity and the change of labels.
> Example:
>
> ## Reasoning type: {E1<->E2}->C
>
> Claim: Bettany Hughes, an English historian scholar, born May 15th, 1967, presented "The Spartans".
> Evidence:
> (E1)The Spartans (documentary): “ The Spartans ” was a 3-part historical documentary series first broadcast on UK terrestrial Channel 4in 2003 , presented by Bettany Hughes .
> (E2)Bettany Hughes: Bettany Hughes ( born May 15 , 1967 ) is an English historian , author , and broadcaster .
> Label: SUPPORTS
> Generate a counterfactual claim:
> “The Spartans” is a documentary presented by Bettany Hughes, an American historian scholar born on March 24, 1980.
>
> ## Reasoning type: E1 <-> E3 <-> E2, E1 -> C <- E2
>
> Claim: Gina Bramhill was born in an Italian village. The 2011 population of the unitary authority area that includes this village was 167,446.
> Evidence:
> (E1)North Lincolnshire: The population of the Unitary Authority at the 2011 census was 167,446 .
> (E2)Gina Bramhill: Gina Bramhill was born in Eastoft , where she grew up on a farm .
> (E3)Eastoft: Eastoft is a village and civil parish in North Lincolnshire , England .
> Label: REFUTES
> Generate a counterfactual claim:
> In 2011, the population of the North Lincolnshire, including an English village where the birthplace of Gina Bramshill is located, was 167,446.
>
> ## Reasoning type: E1 -> C <- E2
>
> Claim: University of Wisconsin Madison is a research university and University of Notre Dame is a research university.
> Evidence:
> (E1)University of Wisconsin–Madison: The University of Wisconsin–Madison ( also known as University of Wisconsin , Wisconsin , UW , or regionally as UW–Madison , or simply Madison ) is a public research university in Madison , Wisconsin , United States .
> (E2)University of Notre Dame: The University of Notre Dame du Lac ( or simply Notre Dame ) is a Catholic research university located adjacent to South Bend , Indiana , in the United States .
> Label: SUPPORTS
> Generate a counterfactual claim:
> The American University of Wisconsin-Madison is a public research university while Notre Dame is not.
>
>
> Claim: The writer Norman Alfred William Lindsay enjoyed boxing, but the author of The Hundred Secret Senses did not.
> Evidence:
> Amy Tan: Amy Tan ( born February 19 , 1952 ) is an American writer whose works explore mother-daughter relationships and the Chinese American experience .
> The Hundred Secret Senses: The Hundred Secret Senses is a bestselling 1995 novel by Chinese-American writer Amy Tan .
> Norman Lindsay: Norman Alfred William Lindsay ( 22 February 1879 – 21 November 1969 ) was an Australian artist , etcher , sculptor , writer , editorial cartoonist , scale modeller , and an accomplished amateur boxer .
> Label: SUPPORTS
> Generate a counterfactual claim:
> ```
> ***Generated counterfactual claim:***
> ```
> The writer Norman Alfred William Lindsay not only enjoyed boxing, but also shared a passion for it with the author of The Hundred Secret Senses, Amy Tan.
> ```

---

### Meta-Review · Area_Chair_Pg4c · 2023-09-19

**Recommendation:** 5

**Metareview:**

The paper presents a data augmentation technique for Multi-hop Fact Verification which generates diverse counterfactuals via a proposed  Explain-Edit-Generate architecture. The experimental results highlights the potential of the approach in generating linguistically diverse counterfactual data while preserving the logical relationships. The writing requires further polishing (starting from the abstract, but this is minor could be addressed in the current version). Several points were clarified during the discussion phase which I strongly encourage the authors to incorporate into their papers.

---

### Decision · Program_Chairs · 2023-10-07

**Decision:**

Accept-Main

**Comment:**

The paper presents a data augmentation technique for Multi-hop Fact Verification which generates diverse counterfactuals via a proposed  Explain-Edit-Generate architecture. The experimental results highlights the potential of the approach in generating linguistically diverse counterfactual data while preserving the logical relationships. The writing requires further polishing (starting from the abstract, but this is minor could be addressed in the current version). Several points were clarified during the discussion phase which I strongly encourage the authors to incorporate into their papers.